# Protective effect of *Platymiscium floribundum* Vog. in tree extract on periodontitis inflammation in rats

**Jordânia M. O. Freire**[1], **Hellíada V. Chaves**[1], **Alrieta H. Teixeira**[2], **Luzia Herminia T. de Sousa**[1], **Isabela Ribeiro Pinto**[1], **José Jackson do N. Costa**[3], **Nayara Alves de Sousa**[4], **Karuza Maria A. Pereira**[5], **Virgínia C. C. Girão**[5], **Vanessa C. S. Ferreira**[6], **João Evangelista de Ávila dos Santos**[7], **Mary Anne S. Lima**[7], **Antônia T. A. Pimenta**[7], **Raquel de C. Montenegro**[8], **Maria Elisabete A. de Moraes**[8], **Vicente de P. T. Pinto**[4], **Gerardo C. Filho**[4], **Mirna M. Bezerra**[4,8] *

**1** Northeast Biotechnology Network–Ph.D. Program, Federal University of Ceará, Sobral, Ceará, Brazil, **2** School of Dentistry, Federal University of Ceará, Sobral, Ceará, Brazil, **3** School of Medicine, University CenterINTA–UNINTA, Sobral, Ceará, Brazil, **4** School of Medicine, Federal University of Ceará, Sobral, Ceará, Brazil, **5** Department of Morphology, School of Medicine, Federal University of Ceará, Fortaleza, Ceará, Brazil, **6** Postgraduate Program in Morphofunctional Sciences, Department of Morphology, School of Medicine, Federal University of Ceará, Fortaleza, Ceará, Brazil, **7** Department of Organic and Inorganic Chemistry, Federal University of Ceará, Fortaleza, Ceará, Brazil, **8** Drug Research and Development Center (NPDM), Federal University of Ceará, Fortaleza, Ceará, Brazil

* mirna@ufc.br

**Data Availability Statement:** All relevant data are within the manuscript and Supporting Information files.

## Abstract

Periodontitis is an immuno-inflammatory disease, which can lead to tooth loss. This study aimed to investigate the efficacy of *Platymiscium floribundum* Vog., a Brazilian tree which has been used in folk medicine as an anti-inflammatory agent, in a pre-clinical trial of periodontitis in rats. Periodontitis was induced by placing a sterilized nylon (3.0) thread ligature around the cervix of the second left upper molar of the rats, which received (*per os*) *P. floribundum* extract (0.1, 1 or 10 mg/kg) or vehicle 1h before periodontitis-challenge and once daily during 11 days. Treatment with *P. floribundum* (10mg/kg) decreased alveolar bone loss, MPO activity nitrite/nitrate levels, oxidative stress, TNF-α, IL1-β, IL-8/CINC-1, and PGE2 gingival levels, and transcription of TNF-α, IL1-β, COX-2, iNOS, RANK, and RANKL genes, while elevated both BALP serum levels and IL-10 gingival levels. The animals did not show signs of toxicity throughout the experimental course. These findings show that *P. floribundum* has anti-inflammatory and anti-resorptive properties in a pre-clinical trial of periodontitis, representing an interesting biotechnological tool.

## Introduction

Periodontitis is a complex immuno-inflammatory disease being characterized by periodontal ligament damage and alveolar bone loss, which can lead ultimately to tooth loss. Its complexity arises from the interplay between tooth-associated microbial biofilm and the host's immune-

**Funding:** This work was supported by Brazilian grants from Conselho Nacional de Pesquisa (CNPq) (grant #471974/2013-7), Coordenação de Aperfeiçoamento de Pessoal de Nível Superior (Capes) and Fundação Cearense de Apoio ao Desenvolvimento Científico e Tecnológico (Funcap) (Capes/Funcap grant # AE1-0052- 000180100/ 2011), and Instituto de Biomedicina do Semi-Árido Brasileiro (INCT-IBISAB). The funders had no role in study design, data collection and analysis, decision to publish, or preparation of the manuscript.

**Competing interests:** The authors have declared that no competing interests exist.

inflammatory response [1]. Due its prevalence, periodontitis has negative effects on the quality of life of the subjects.

Within inflamed periodontium a plethora of cells (epithelial cells, fibroblasts, leukocytes, osteoblasts, osteoclasts, and dendritic cells) release inflammatory mediators such as cytokines/ chemokines (TNF-α, IL1-β, IL-8/CINC-1), as well as prostaglandins (PGE2), and nitric oxide (NO), which promote the breakdown of the tooth supporting tissues [2]. Also, it is worth mentioning that reactive oxygen species (ROS) can also play a role in this process [3].

The conventional treatment for periodontitis comprises the non-surgical (scaling and root planning) and surgical approaches. Furthemore, adjunct therapy with systemic nonsteroidal anti-inflammatory drugs and antibiotics is sometimes required [4]. However, this strategy has attracted a lot of criticisms because the side effects such as emerging of bacterial resistance, gastrointestinal bleeding as well as cardiovascular and renal effects [5]. In an attempt to overcome these limitations the search for safer therapeutic agents still continues.

Natural products isolated from plants are considered good alternatives to synthetic chemicals. *Platymiscium floribundum* Vog. (genus Platymiscium and family Fabaceae-Papilionoideae) is a medium sized tree which belongs to the Fabaceae-Papilionoideae family. In the Brazilian Northeast it is popularly known as "sacambu" and "jacaranda-do-litoral" and it has been used in folk medicine as an anti-inflammatory agent. Phytochemical investigation of this genus highlighted flavonoids, isoflavones and coumarins as main constituents [6–7].

The use of natural products in the management of periodontitis still lacks preclinical and clinical studies that can prove the efficacy and safety. Thus, the present study was aimed at investigating the unexplored efficacy of *P. floribundum* in a rat model of periodontitis. Also, we investigated the putative role of cytokines/chemokines PGE$_2$, NO, and oxidative stress in *P. floribundum* efficacy. Additional, a systemic evaluation of the sub-chronic toxicity of *P. floribundum* was carried out.

## Materials and methods

### Animals

144 female Wistar rats (200–220g) were obtained from the Federal University of Ceara animal research facility. Rats were housed in polypropylene boxes with controlled temperature and 12 h-12 h light dark cycles with free acess to chow diet and water. All animal procedures followed a protocol in compliance with the guidelines from the Brazilian Society of Laboratory Animal Science (SBCAL) and which was approved by a standing Institutional Animal Care and Use Committee at School of Medicine, Federal University of Ceará, Sobral, Ceará, Brazil (Permit number: 05/2015).

### Plant material

*P. floribundum* was collected in Acarape, Ceará State, Brazil. A voucher specimen (#31052) identified by Prof. Edson Paula Nunes is deposited at the Prisco Bezerra Herbarium, Federal University of Ceará, Brazil [8]. For this study we used part of the chloroformic extract (2.0 g) obtained from trunk heartwood (8,000.0 g) of *P. floribundum*.

### Experimental protocol

Animals were divided into unchallenged and periodontitis-challenged groups (6 rats each) receiving (*per os*) either *P. floribundum* (0.1, 1 or 10 mg/kg) or vehicle (0.9% saline + 0.1% ethanol) 1h before periodontits-challenge and once daily during 11 days.

Periodontitis-challenge was performed, under anesthesia (i.p) (ketamine 90 mg/kg + xylazine 10 mg/kg), by placing a nylon thread (3.0 Nylpoint, Ceará, Brazil) around the second molar [9]. On the 11th day, rats were euthanized with an overdose of ketamine/xylazine (300:30mg/kg; i.p.). The maxillae were harvested to analyze the bone loss. *P. floribundum* 10 mg/kg was found to be the most effective dose at protecting against alveolar bone loss, and therefore this dose was chosen for hystopathological analyses, and for the quantification in gingival tissues of (1) Myeloperoxidase (MPO) activity, (2) nitrite/nitrate levels, (3) superoxide dismutase-SOD/catalase-CAT levels, (4) TNF-α, IL1-β, IL-8/CINC-1, IL-10, and PGE2 levels (ELISA), and (5) qRT-PCR for TNF-α, IL1-β, COX-2, iNOS, RANK and RANKL. All dosages were done on the 11$^{th}$ day, except for the MPO dosage that was done at the 6$^{th}$ hour.

## Determination of bone remodeling

After 11 days of treatment, the maxillae were harvested and fixed in buffered formalin (10%). After 24 hour, the maxillas were defleshed and stained with methylene blue (1%), fixed in a piece of wax, and photographed. The morphometric analysis of bone resorption was performed using the ImageJ® Software (National Institute of Health, Bethesda, MD, USA), as described previously [10]. Further, serum levels of Bone Alkaline Phosphatase (BALP) were quantified as previously described [11] for analysis of bone formation.

## Histopathological analysis of alveolar bone

The H&E-stained maxillae were semiquantitatively evaluated for the presence of cell infiltrate and osteoclasts, and state of preservation of cement and alveolar process. giving a (0–3) score grade for each of these parameters [9].

## Scanning Electron Microscopic (SEM) of alveolar bone

The maxillae were fixed in Karnovisky for at least 6 hours, and then transferred to Cacodylate buffer. They were then cut into a diamond-shaped blade in a medial-distal plane to obtain the maxillary fragment (0.5 × 0.2 cm and 0.5 mm thick). The fragments were left in desiccator drying for 24 h and were assembled into gold dust plating stubs (Quorum Metallizer QT150ES, Quorum Technologies, Laughton, England) for Scanning Electron Microscopy (SEM inspect50, FEI, Hillsboro, Oregon, USA). The analysed region was that between the first and second molar.

## MPO activity

MPO activity was analyzed in gingival tissues collected at 6$^{th}$ hour of periodontitis-challenge, as described by Bradley et al. [12]. The results were expressed as MPO activity/mg tissue.

## NO production

The total nitrite/nitrate dosage was performed as an indicator of NO production in gingival samples by Griess reaction [13]. The values obtained were compared with those obtained for standard curve and expressed in NOx (μM).

## SOD and CAT activity assessment

SOD was measured using a protocol previously described [14]. The results are expressed in grams of SOD/mL. CAT activity was performed using a protocol described by Maehly and Chance [15]

## Cytokines and PGE2 dosage

TNF-α, IL-1β, IL-8/CINC-1, and IL-10 levels were evaluated in gingival samples using commercial kits DuoSet ELISA (R&D Systems Inc., MN, USA). In the same way, the determination of PGE2 was performed by ELISA, using R&D Systems®, Kit Parameter TMPGE2 Assay (catalog PKGE004B, USA kit). Both kits were used according to manufacturer's instructions. The results were arranged as pg/mL.

## RNA isolation and quantitative real-time PCR

Total RNA was extracted from gingival samples using the TRIzol reagent (Invitrogen, São Paulo, Brazil). The reverse transcription was performed using SuperScript IV Reverse Transcriptase (Invitrogen, São Paulo, Brazil) following the manufacturer's instructions.

Quantitative real-time polymerase chain reaction (qRT-PCR) was performed in StepOne Real-Time PCR thermocycler (Applied Biosystems,Warrington, UK) using SYBRR Green Master Mix (Applied Biosystems, Warrington, UK), as indicated by the manufacturer. The relative gene expression was determined using the $2^{-\Delta\Delta Ct}$ method [16], with GAPDH (*S–GGACC AGGTTGTCTCCTGTG/A–CATTGAGAGCAATGCCAGCC*) as the housekeeping gene. The primers pairs used in this study were: TNF-α (*S–CGGGGTGATCGGTCCCAACAAG/ A–GT GGTTTGCTACGACGTGGGC*), IL1-β (*S–TGCTGTCTGACCCATGTGAG/ A–CCAAGGCC ACAGGGATTTTG*), COX-2 (*S–TCCAGTATCAGAACCGCATTGCCT/A AGCAAGTCCGTG TTCAAGGAGGAT*), iNOS (*S–AGGCACAAGACTCTGACACC/ A–GGTAGGGTAGAGGAG GGGAG*), RANK (*S–AGGGAAAACGCTGACAGCTAA/ A–CCAACACAATGGTCCCCTGA*), and RANKL (*S–GCCAACCGAGACTACGGCAA / A–GAACATGAAGCGGGAGGCG*).

## Toxicity assessment

After 11 days of treatment, rats were anesthetized and blood samples were collected from the right ventricle and centrifuged for dosing of total alkaline phosphatase (TALP), creatinine, alanine aminotransferase (ALT) and aspartate aminotransferase (AST). Dosages were done following the manufacturer's instructions (Labtest®, Lagoa Santa, MG, Brazil). Additionally, after euthanasia, heart, stomach, liver and kidneys were removed for histopathological analysis (H&E).

## Statistical analysis

All data were normalized using the Shapiro-Wilk normality test. Results were showed as mean ±standard error (SEM) or as median, when appropriate. ANOVA followed by the Tukey test or Games-Howell test were used to compare the means and the Kruskal-Wallis and Dunn tests were used to compare the medians. P <0.05 was considered significant. Analyzes were performed using IBM SPSS Statistics for Windows, Version 20.0. Software Armonk, NY or GraphPad Prism 6, San Diego, CA, USA.

# Results

## Effect of *P. floribundum* on alveolar bone loss

Administration of *P. floribundum* (10 mg/kg) 1 hour before the placement of the ligature and for 11 days resulted in a significant inhibition (p<0.001) of alveolar bone loss (Fig 1A), compared to the groups that received only the vehicle (non-treated group). Periodontitis-challenge was associated with alveolar bone resorption, root exposure and loss of interdental contact (Fig 1B [b]), when compared to unchallenged group (Fig 1B [a]). These findings were confirmed by the histopathological analysis (H&E), which showed an intense inflammatory cell

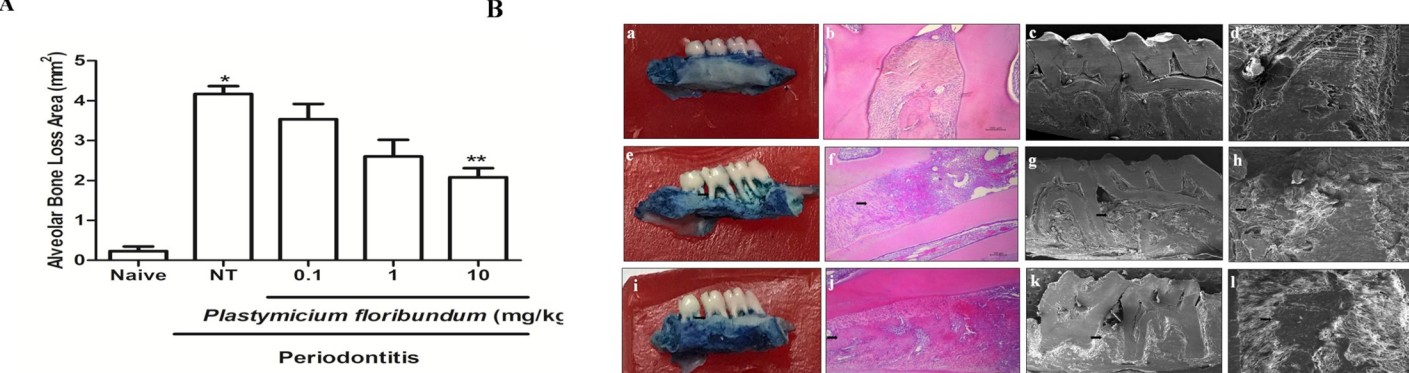

**Fig 1. (A) Effect of *P. floribundum* (0.1; 1 or 10 mg/kg) on alveolar bone loss in experimental periodontitis.** Naive: animals without periodontitis; NT: animals subjected to periodontitis and treated with vehicle (0.9% saline + 0.1% ethanol). *P. floribundum*: rats submitted to periodontitis and treated with *P. floribundum* (0.1; 1 or 10 mg/kg), respectively. Data are shown as mean ± SEM (n = 6 for each treatment). *P < 0.000039 *versus* naive; **P < 0.001 *versus* NT, (ANOVA; Games-Howell).
**(B) Effect of *P. floribundum* (0.1; 1 or 10 mg/kg) on the macroscopic view (first column), histological aspects (second column), and Scanning Electron Microscopy (SEM) (third columm) of periodontium.** (a–d) normal maxilla (naive), showing the integrity of its components (C—Cementum; D–Dentin, and AB—Alveolar Bone). (e–h) maxilla from rats subjected to periodontitis and receiving only vehicle (0.9% saline + 0.1% ethanol) presenting severe bone resorption, inflammatory infiltrate in the gingiva and periodontal ligament, extensive destruction of cementum, total resorption of the alveolar process (f), and irregularity in the bone tissue (g—h). (i–l) maxilla from rats subjected to periodontitis and treated with *P. floribundum* (10 mg/kg) showing discrete cellular influx and preservation of cementum and alveolar process (j) and regular tissue topography (k–l). Black arrows indicate alveolar bone resorption. HE magnification (100x); MEV magnification 65x: c, g, and k; MEV magnification 350x: d, h, and i.

influx, reabsorption of the alveolar bone with partial destruction of the cementum and presence of osteoclasts (Fig 1B [f]), receiving median scores of 2 (2–3) (Table 1). *P. floribundum* (10 mg/kg) decreased root exposition and loss of interdental contact (Fig 1B [i]), when compared to non-treated group (Fig 1B [b]). Also, the histopathological analysis (H&E) depicted a discrete inflammatory cellular infiltrate, partial preservation of the alveolar process, preserved cement, and reduction of osteoclasts number (Fig 1B [j]), receiving scores from 0–1 (Table 1). These values were statistically different (P < 0.001), when compared with the non-treated group. When analyzing the region between the first and second molar by using Scanning Electron Microscopy (SEM) in both 65x and 350x magnification, respectively, it can be seen that periodontitis-challenge was associated with an irregular topography (Fig 1B [g-h]), when compared to the bone tissue from unchallenged group (Fig 1B [c-d]). The rats treated with *P. floribundum* (10 mg/kg) (Fig 1B [k-1]) showed regular tissue topography, when compared to non-treated group (Fig 1B [g-h]).

Also, periodontitis-challenge was associated with a decrease in BALP levels, which was prevented by the *P. floribundum* (10 mg/kg) injection, when compared with the non-treated group (Fig 1C).

**Table 1. Effect of *P. floribundum* on histopathology (H&E) in the maxilla from rats subjected to periodontitis.**

| | | | | *P. floribundum* | |
| --- | --- | --- | --- | --- | --- |
| | **Naive** | **NT** | **0.1** | **1** | **10** |
| **Median and Variation** | 0 (0–0) | 2 (2–3) * | 2 (1–3)* | 1 (0–2)* | 0 (0–1)** |

*P <0.001 versus naive

**P <0.001 *versus* non-treated group (NT) (Kruskal-Wallis; post hoc Dunn's Test).

### Effect of *P. floribundum* on MPO activity

A subset of rats was euthanized at the 6th hour for analysis of MPO activity, where it was found a significant increase in MPO activity, when compared to naive group (Fig 2). *P. floribundum* (10 mg/kg) decreased MPO activity, when compared with to non-treated group (Fig 3).

Periodontitis-challenged group displayed a significant increase in nitrite/nitrate levels ($P < 0.05$) (Fig 4A). Also, it was found a significant reduction in SOD ($P<0.0003$) and CAT ($P < 0.007$) levels, when compared to naive group (Fig 4B and 4C). *P. floribundum* (10 mg/kg) decreased nitrite/nitrate levels ($P < 0.0317$) (Fig 4A), and it increased both markers for oxidative stress (SOD and CAT) ($P < 0.013$ and $P < 0.002$, respectively) (Fig 4B and 4C), when compared with to non-treated group.

### Effects of *P. floribundum* on both cytokines and PGE2 levels

Periodontits-challenge was associated with increased levels of TNF-α ($P < 0.00001$), IL-1β ($P < 0.000001$), and IL-8/CINC-1 ($P < 0.017$), when compared to the naive group (Fig 5A–5C). Still, it was observed a significant ($P < 0.01$) decrease in gingival levels of IL-10, when compared to naive group (Fig 5D). Also, it was found significantly ($P<0.0001$) higher gingival PGE-2 levels in non-treated rats, when compared to the naive group. *P. floribundum* (10 mg/kg) injection decreased the levels of both proinflammatory cytokines and PGE2, and increased IL-10 levels (Fig 5E).

### Effects of *P. floribundum* on the mRNA levels of TNF-α, IL-1β, COX-2, iNOS, RANK, and RANK-L

Periodontitis-challenged group expressed a significant increase in the mRNA levels of TNF-α ($P<0.001$), IL-1β ($P<0.0000001$), COX-2 ($P<0.000004$), iNOS ($p< 0.007$), RANK ($P<0.036$),

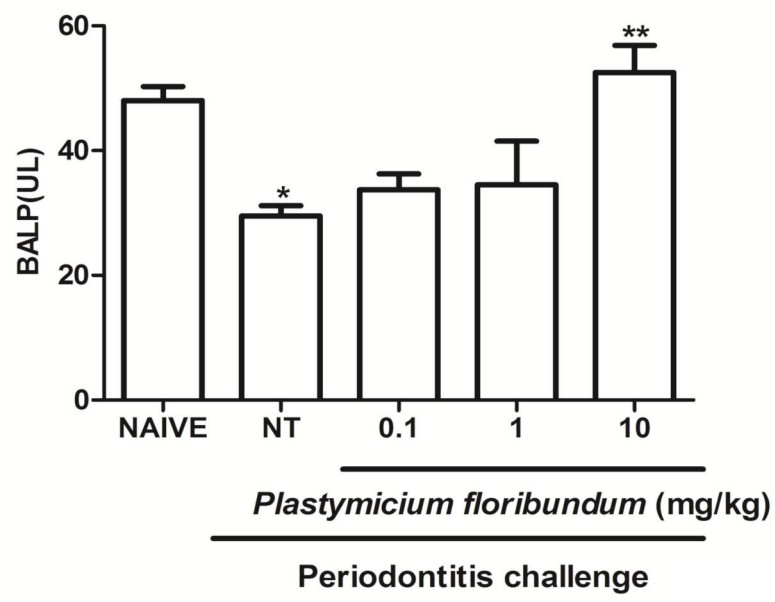

**Fig 2. Effect of *P. floribundum* on the serum alkaline phosphatase levels.** Data represent the mean ± SEM of six animals for each group. *p<0.001 *versus* naive; **p<0.012 versus non-tretaed (NT) (ANOVA; post hoc Games-Howell).

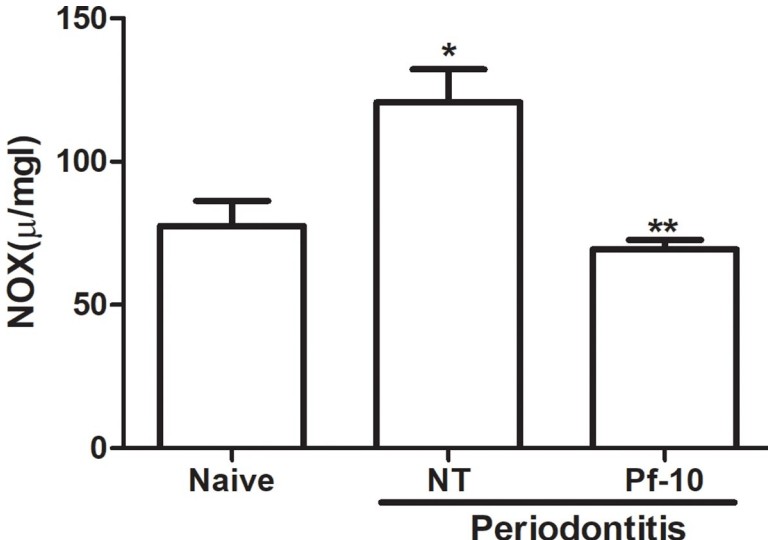

**Fig 3. Effect of *P. floribundum* (10mg/kg) on gingival myeloperoxidase (MPO) activity.** Data are shown as mean ± SEM (n = 6 for each treatment). *P < 0.000023 versus naive; **P < 0.01046 versus non-treated NT, (ANOVA; Tukey).

and RANK-L (P<0.05), when compared to naive group (Fig 6A–6F). *P. floribundum* (10 mg/kg) injection significantly reduced the mRNA levels of TNF-α, IL-1β, COX-2, iNOS, RANK (P<0.036), and RANK-L, when compared to the vehicle group (P < 0.001; P < 0.0000001; P < 0.000004; P < 0.001; P < 0.039; and P < 0.036, respectively) (Fig 6A–6F).

### Evaluation of *P. floribundum* toxicity

During the 11 days of treatment with *P. floribundum* the animals do not showed any signs of toxicity or mortality. The serum levels of Total Alkaline Phosphatase (TALP), creatinine, as well as serum levels of liver enzymes (AST and ALT) were unchanged, when compared to non-treated animals (Table 2).

The histopathological analysis of the liver and kidney from animals treated with *P. floribundum* (10 mg/kg) depicted reversible and harmless changes, including: mild cellular swelling, few haemorrhagic areas, and areas of discrete congestion (Fig 7D and 7F), when compared to the vehicle treated group. However, all these histological changes are considered reversible. The heart and gastric mucosa of the animals did not show any histopathological changes (Fig 7B–7H).

### Discussion

Herein we have demonstrated the anti-inflammatory and anti-resorptive effect of *P. floribundum* in a ligature-induced periodontitis in rats. In this study *P. floribundum* (10 mg/kg) reduced alveolar bone loss and increased the serum levels of BALP, an isoenzyme of alkaline phosphatase considered a marker of bone formation. Furthermore, *P. floribundum* (10mg/kg) decreased MPO activity, TNF-α, IL1-β, IL-8/CINC-1, and PGE2 gingival levels, oxidative stress, and transcription of TNF-α, IL1-β, COX-2, iNOS, RANK and RANKL genes, while elevated IL-10 gingival levels. Additional, the animals did not show signs of toxicity throughout the experimental course.

Several species of Fabaceae family shows pharmacological activities, thus being a promising source of new bioactive products. Many of these plants are used in popular medicine showing

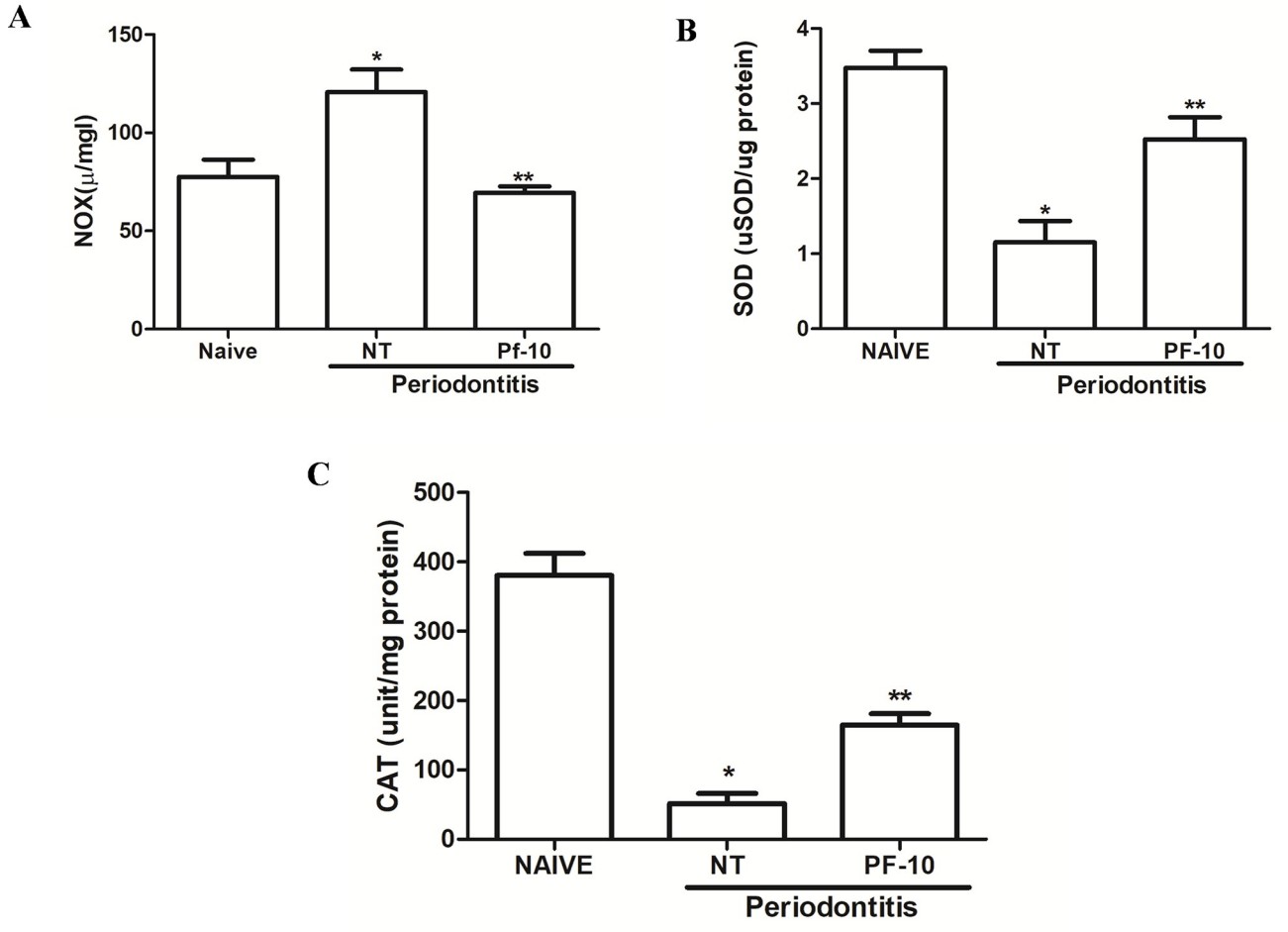

**Fig 4.** (A) **Effect of *P. floribundum* (10mg/kg) on gingival nitrite/nitrate levels (NOx).** Data are shown as mean ± SEM (n = 6 for each treatment). *P < 0. 0015 versus naive; **P < 0. 0317 *versus* non-treated (NT) (ANOVA; Tukey). (B) **Effect of *P. floribundum* (10mg/kg) on gingival SOD (B) and CAT (C) levels.** Data are shown as mean ± SEM (n = 6 for each treatment). * P < 0.0003 *versus* naive; ** p< 0.013 *versus* NT (ANOVA; Tukey). *P < 0.007 *versus* naive; **P < 0.002 *versus* non-treated (NT) (ANOVA; Games-Howell).

effects such as expectorant, analgesic, treatment of asthma and abdominal pain [17–18]. Although *P. floribundum* is popularly used as an anti-inflammatory [19], the literature has few studies concerning its mechanism of action. In this regard, phytochemical investigations demonstrated the induction of apoptosis by flavonoids from *P. floribundum* in HL-60 human leukemia cells [6]. Further, it was reported *P. floribundum* antifungal, DNA-damaging and anticholinesterasic activities [7]. The use of natural products in the management of periodontitis still lacks preclinical and clinical studies that can prove its efficacy and safety. *P. floribundum* was chosen to be studied because it has been used in the Brazilian Northeast in folk medicine as an anti-inflammatory agent and no previous pharmacology study was reported yet. Indeeed, the present study is the first demonstration of both efficacy and safety of *P. floribundum* as anti-inflammatory agent.

BALP has been implicated in bone formation [20]. In the present study, *P. floribundum* (10 mg/kg) increased serum levels of BALP, which suggests that it may promote bone formation and prevent alveolar bone loss. *P. floribundum* have some secondary metabolites such as

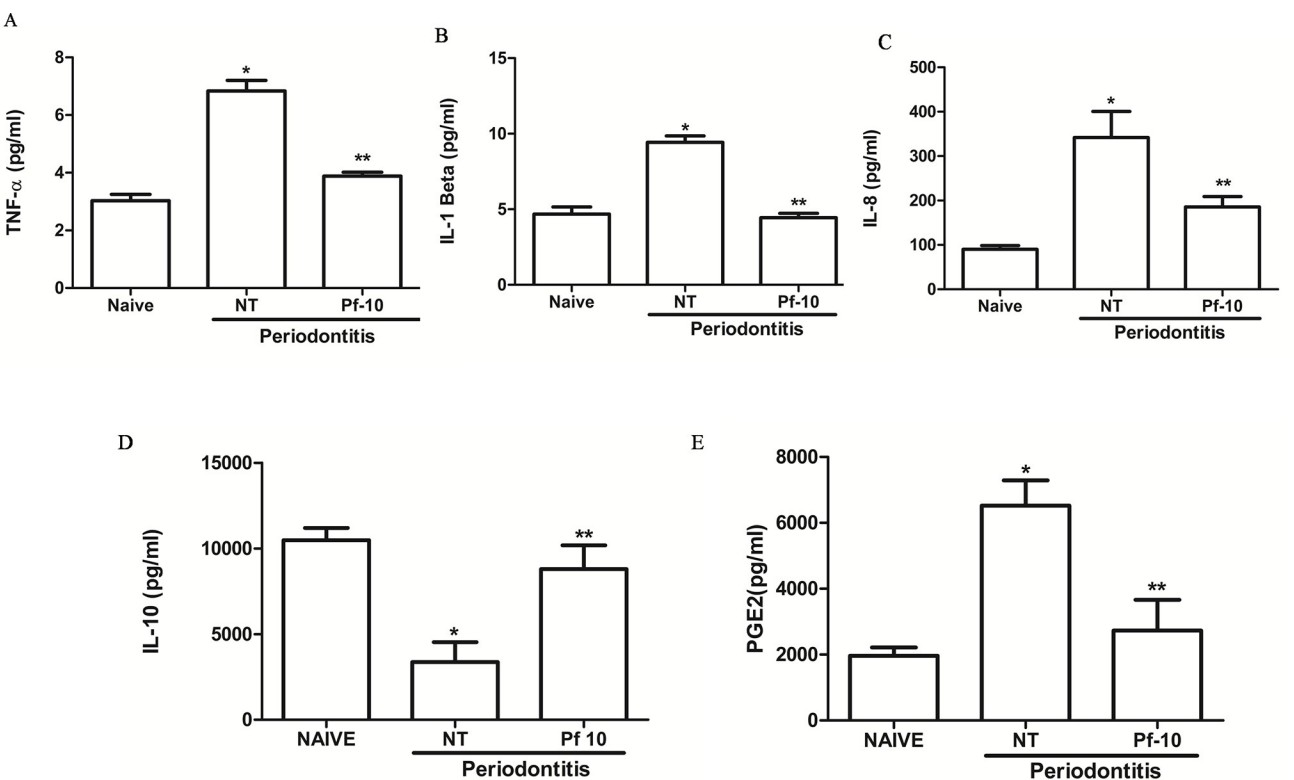

**Fig 5. Effects of *P. floribundum* on TNF-α (A), IL-1β (B), IL-8/CINC-1 (C), IL-10 (D), and PGE2 levels in gingival tissues.** Data are shown as mean ± SEM (n = 6 for each treatment). *P < 0.00001 *versus* Naive and **P < 0.001 *versus* NT for TNF-α; *P < 0.017 *versus* Naive and **P < 0.05 *versus* NT for IL-8; *P < 0.000001 *versus* Naive and **P < 0.0000001 *versus* NT for IL-1β; *P < 0.000 *versus* Naive and **P < 0.004 *versus* NT for PGE2; *P < 0,01 *versus* Naive and **P < 0.02 *versus* NT for IL-10 (ANOVA; Games-Howell; post hoc Tukey).

flavonoids, isoflavones, and coumarins [7], and some authors have shown that these metabolites are able to decrease inflammatory mediators during bone resorption [21].

MPO activity was used as a marker of inflammatory cell infiltrates (especially neutrophils) in the inflamed periodontium. According to Palm et al. [22], MPO may be used as a biomarker of periodontitis and high saliva MPO concentrations may suggest periodontal tissue destruction. In the present study, periodontitis-challenge was associated with a significant increase in MPO activity. *P. floribundum* (10 mg/kg) decreased MPO acivity, suggesting that the efficacy of *P. floribundum* may be associated with reduction of neutrophil influx.

The periodontitis-associated microbiota can activate resident and recruited inflammatory cells to secrete proinflammatory cytokines/chemokines, which stimulate bone resorption by stimulating the activity of osteoclasts [23]. Indeed, these cytokines increase the recruitment and activity of these cells through improved production of a key osteoclastogenic factor, the Receptor Activator of Nuclear Factor κ B Ligand (RANK-L) and favor bone damage [2]. Also, some authors have suggested the potential involvement of both pro- and anti-inflammatory cytokines in the regulation of the chronic inflammatory periodontitis. In fact, our research group recently showed that ligature-induced periodontitis in rats was related to a significant decrease in gingival levels of IL-10, an anti-inflammatory cytokine [24] In the preset study, *P. floribundum* (10mg/kg) decreased TNF-α, IL1-β, and IL-8/CINC-1 gingival levels and transcription of TNF-α, IL1-β, RANK and RANKL genes, while elevated IL-10 gingival levels,

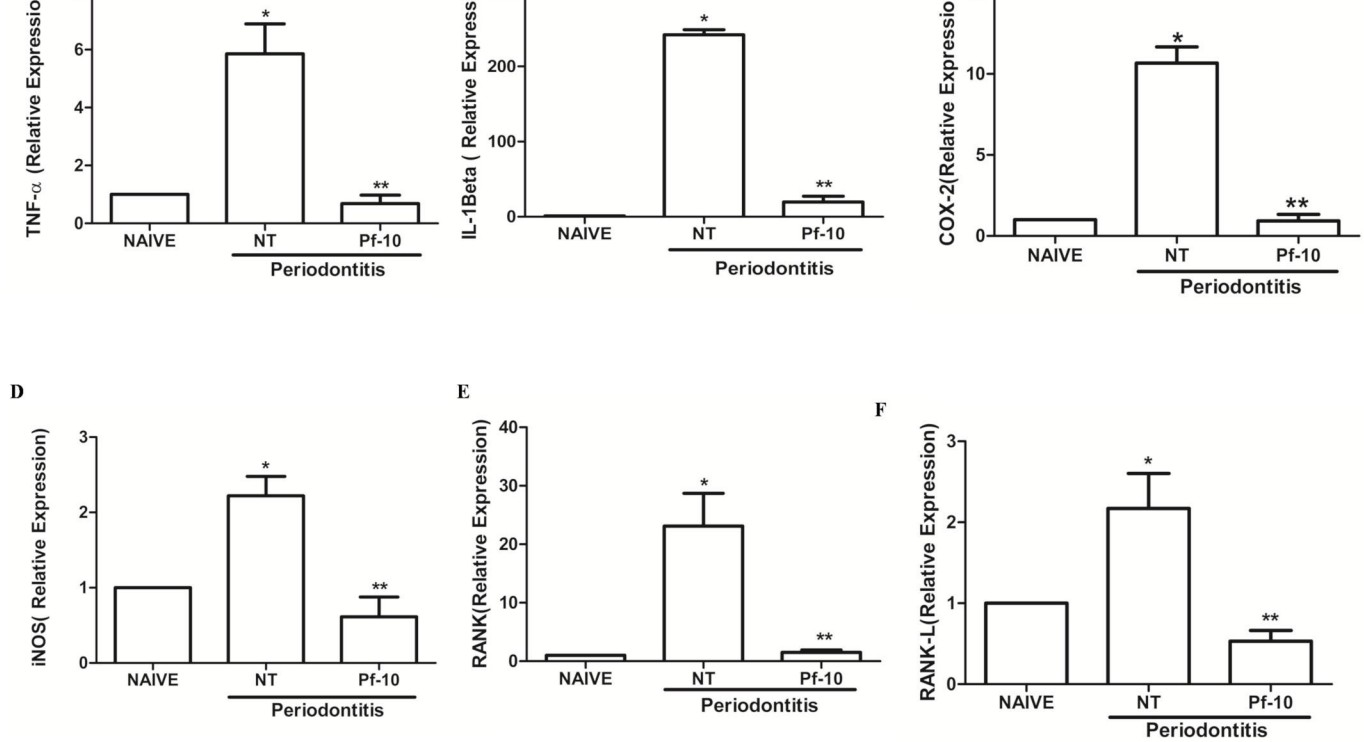

**Fig 6. Effect of *P. floribundum* (10mg/kg) on the mRNA levels of TNF-α, IL-1β, COX-2, iNOS, RANK, and RANK-L in gingival tissues from rats subjected to periodontitis.** Data are shown as mean ± SEM (n = 6 for each treatment). **A.** * P < 0.001 *versus* Naive; **P < 0.001 versus NT. **B.** *P < 0.0000001 *versus* Naive; **P < 0.0000001 *versus* NT. **C.** * P < 0.000004 *versus* Naive; **P < 0.000004 *versus* NT. **D.** * P < 0.007 *versus* Naive; ** P < 0.001 *versus* NT. **E.** *P < 0.036 *versus* Naive; ** P < 0.039 *versus* NT. **F.** * P < 0.05 *versus* Naive; ** P < 0.036 *versus* NT (ANOVA; Tukey; Games-Howell, respectively).

which suggests that the protective effect of *P. floribundum* may be related to the modulation of both pro- and anti-inflammatory cytokines levels in gingival tissue.

The biosynthesis of PGE2 is close related to pro-inflammatory cytokines activity. Some authors showed that the biosynthesis of PGE2 is increased in inflamed gingiva and this synthesis is enhanced by pro-inflammatory cytokines (TNF-α, IL-1β), creating a vicious circle in states of inflammatory osteolysis, promoting alveolar bone loss [2]. Sánchezz et al. [25] suggested the salivary levels of both IL-1β and PGE2 as biomarkers of periodontal status. Our research group recently demonstrated that periodontitis-challenge in rats is associated with both high gingival PGE-2 levels and increased periodontal immunostaining for its processing

**Table 2. Evaluation of treatment with *P. floribundum* on biochemical parameters in rats submitted to induction of periodontitis.**

| Parameters | Naive | NT | 0.1 | *P. floribundum* 1 | 10 |
|---|---|---|---|---|---|
| Total Alkaline Phosphatase | 87.86 ± 9.79 | 84.54 ± 6.62 | 87.30 ± 7.40 | 64.93 ± 4.32 | 65.60 ± 4.96 |
| Creatinine | 11.56 ± 0.26 | 12.73 ± 0.55 | 12.60 ± 0.67 | 13.60 ± 0.24 | 13.72 ± 0.68 |
| AST (U/I) | 114.4 ± 7.262 | 123.8 ± 7.667 | 103.6 ± 3.125 | 99.93 ± 4.581 | 102.2 ±2.615 |
| ALT (U/I) | 38.67 ± 2.539 | 33.46 ± 0.8512 | 40.00 ± 3.36 | 37.50 ± 2.61 | 38.38 ± 1.840 |

Data represent the mean ± S.E.M. (ANOVA; Tukey's).

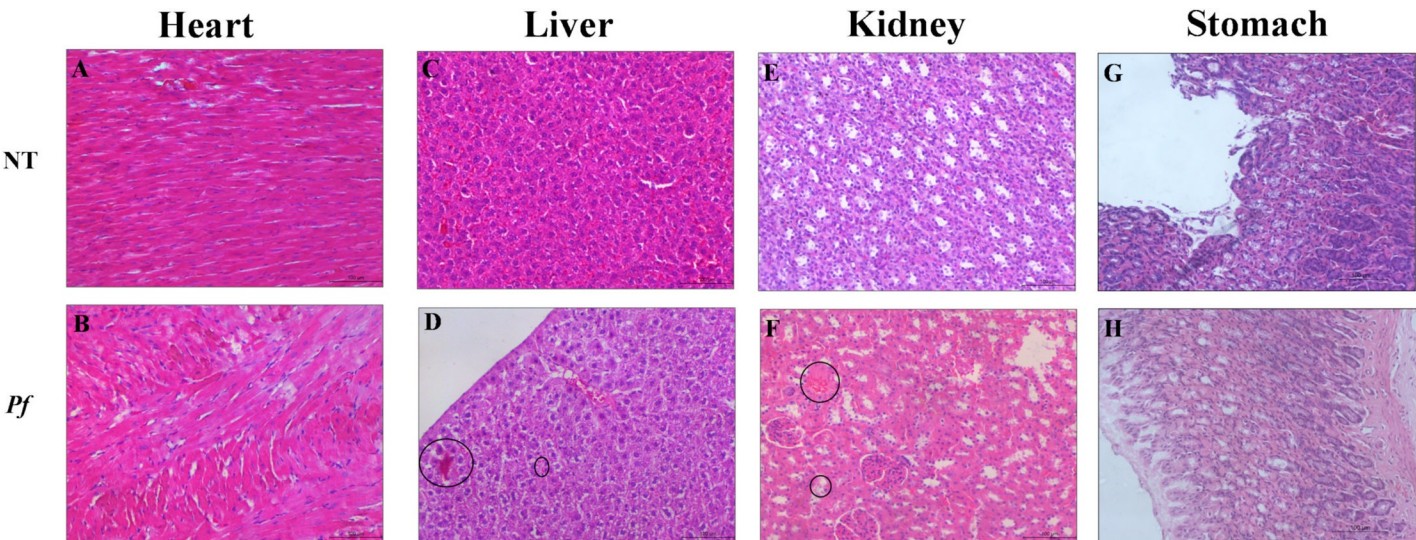

**Fig 7. Photomicrographs of organs from rats subjected to periodontitis and treated with *P. floribundum* (10mg/kg). (A)** Heart, **(C)** Liver, **(E)** Kidney, and **(G)** Stomach from non-treated (NT) group (rats receiving only vehicle). **(B)** Heart, **(D)** Liver, **(F)** Kidney, and **(H)** Stomach from rats treated *P. floribundum* (10mg/kg). Black circles indicate cellular swelling, few hemorrhagic areas (D), areas of discrete congestion (F). Magnification 100x.

enzyme COX-2 [26]. In the present study, we demonstrated that ligature-induced periodontitis rats expressed a significant increase in the mRNA of COX-2. *P. floribundum* reduced both PGE2 levels and the transcription of COX-2 genes in inflamed gingival tissues, which suggests that its antiresorptive effects are, at least in part, related to inhibition of COX-2.

Both preclinical and clinical trials support the hypothesis that the inhibition of prostaglandins with nonsteroidal anti-inflammatory drugs (NSAIDs) slows periodontitis progression [27]. Once selective COX-2 inhibitors had a better risk-benefit ratio than the standard NSAIDs, we performed the histophatological analysis of the gastric mucosa of *P. floribundum* treated-rats, which showed no evidence of macroscopic and microscope alterations. Our present data is significant considering that gastric mucosal damage is a common side effect of NSAIDs.

Since NO has been shown to have dual effects during inflammatory processes, it would not be different in the periodontitis. While some authors suggested that NO plays an important role in the destruction of periodontal tissues [28], Martins et al. [29] demonstrated that NO donors decreased inflammation and bone loss in a model of periodontits in rats. Pro-inflammatory cytokines participate in the amplification of the inflammatory response during periodontitis disease by the transcription of the inducible nitric oxide synthase (iNOS) that results in the production of high amounts of NO [30] In the present study, periodontitis-challenged rats displayed a significant increase in nitrite/nitrate levels and expressed a significant increase in the mRNA of iNOS in gingival tissue, which was reduced by the treatment with *P. floribundum* (10 mg/kg).

In order to keep the ROS levels in a normal range, enzymatic mechanisms such as SOD and CAT are activated continuously. Here we demonstrated that rats with periodontitis showed a significant reduction in both SOD and CAT levels, when compared to naive rats. *P. floribundum* increased both markers for oxidative stress. These data are in accordance with previous results of our group showing the antioxidant effect of natural extracts from *Stemodia maritima* L. and *Calendula offinalis* [24], [31]

Considering the growing interest in natural products, research concerning not only the efficacy but also the toxicity profile of these products is very encouraged [32–33]. Hence, in the present study, we performed biochemical analysis in peripheral blood from rats 11 days after daily injection of *P. floribundum*. Further, we evaluated the integrity (H&E) of heart, liver and kidneys from rats that received *P. floribundum*. Neither biochemical analysis nor the histopathological ones revealed any signs of toxicity.

In conclusion, *P. floribundum* reduces inflammatory markers associated with periodontitis, such as bone loss, pro-inflammatory cytokines, and oxidative stress. Our findings suggest that this natural product-based, may act in different pathways involved in inflammatory process during periodontitis. Thus, *P. floribundum* may offer a huge of possibilities for planning a supportive periodontal therapy, being promising to the oral health care industry.

## Supporting information

**S1 Dataset. Dataset from Fig 1.**
(RAR)

**S2 Dataset. Data set from Fig 2.**
(RAR)

**S3 Dataset. Dataset from Fig 3.**
(RAR)

**S4 Dataset. Dataset from Fig 4.**
(RAR)

**S5 Dataset. Dataset from Fig 5.**
(RAR)

**S6 Dataset. Dataset from Fig 6.**
(RAR)

**S7 Dataset. Dataset from Table 1.**
(PZF)

**S8 Dataset. Dataset from Table 2.**
(PZF)

## Acknowledgments

The authors thank Anderson Weiny Barbalho Silva and Alana Nogueira Godinho for excellent technical support. We also gratefully thank the Analylitc Central of Federal University of Ceará, Ceará, Brazil, for SEM analysis.

## Author Contributions

**Conceptualization:** Jordânia M. O. Freire, Hellíada V. Chaves, Alrieta H. Teixeira, Luzia Herminia T. de Sousa, Isabela Ribeiro Pinto, Nayara Alves de Sousa, Antônia T. A. Pimenta, Vicente de P. T. Pinto, Mirna M. Bezerra.

**Data curation:** Jordânia M. O. Freire, Hellíada V. Chaves, Mirna M. Bezerra.

**Formal analysis:** Jordânia M. O. Freire, Hellíada V. Chaves, Alrieta H. Teixeira, Isabela Ribeiro Pinto, José Jackson do N. Costa, Nayara Alves de Sousa, Karuza Maria A. Pereira,

Vanessa C. S. Ferreira, Mary Anne S. Lima, Antônia T. A. Pimenta, Vicente de P. T. Pinto, Mirna M. Bezerra.

**Funding acquisition:** Hellíada V. Chaves, Virgínia C. C. Girão, Mary Anne S. Lima, Raquel de C. Montenegro, Maria Elisabete A. de Moraes, Vicente de P. T. Pinto, Gerardo C. Filho, Mirna M. Bezerra.

**Investigation:** Jordânia M. O. Freire, Hellíada V. Chaves, Alrieta H. Teixeira, Luzia Herminia T. de Sousa, Nayara Alves de Sousa, Vanessa C. S. Ferreira, Antônia T. A. Pimenta, Mirna M. Bezerra.

**Methodology:** Jordânia M. O. Freire, Hellíada V. Chaves, Alrieta H. Teixeira, Luzia Herminia T. de Sousa, Isabela Ribeiro Pinto, José Jackson do N. Costa, Nayara Alves de Sousa, Karuza Maria A. Pereira, Virgínia C. C. Girão, Vanessa C. S. Ferreira, João Evangelista de Ávila dos Santos, Mary Anne S. Lima, Antônia T. A. Pimenta, Raquel de C. Montenegro, Maria Elisabete A. de Moraes, Vicente de P. T. Pinto, Mirna M. Bezerra.

**Project administration:** Jordânia M. O. Freire, Hellíada V. Chaves, Vicente de P. T. Pinto, Mirna M. Bezerra.

**Resources:** Hellíada V. Chaves.

**Software:** Jordânia M. O. Freire, Hellíada V. Chaves.

**Supervision:** Jordânia M. O. Freire, Hellíada V. Chaves, Mirna M. Bezerra.

**Validation:** Jordânia M. O. Freire, Hellíada V. Chaves, Mirna M. Bezerra.

**Visualization:** Jordânia M. O. Freire, Hellíada V. Chaves, Mirna M. Bezerra.

**Writing – original draft:** Jordânia M. O. Freire, Hellíada V. Chaves, Mirna M. Bezerra.

**Writing – review & editing:** Jordânia M. O. Freire, Hellíada V. Chaves, Mirna M. Bezerra.

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
