## [Decision Letter · Decision Letter 0]

5 Aug 2019

PONE-D-19-17659

Protective effect of the extract from Platymiscium floribundum Vogin a pre-clinical trial of periodontitis in rats is based on its anti-resorptive, antioxidant, and anti-inflammatory properties

PLOS ONE

Dear Dra Freire,

Thank you for submitting your manuscript to PLOS ONE. After careful consideration, we feel that it has merit but does not fully meet PLOS ONE’s publication criteria as it currently stands. Therefore, we invite you to submit a revised version of the manuscript that addresses the points raised during the review process.

ACADEMIC EDITOR: The manuscript is reviewed at the editorial level and by an expert reviewer. Though, they have identified the significance of the study, however suggested several changes to the manuscript content to improve the presentation quality.  The authors should show bone loss in periodontitis and treatment conditions clearly. Additional editor remarks and comments from a reviewer are noted below for authors response carefully for further consideration.

We would appreciate receiving your revised manuscript by Sep 19 2019 11:59PM. To enhance the reproducibility of your results, we recommend that if applicable you deposit your laboratory protocols in protocols.io, where a protocol can be assigned its own identifier (DOI) such that it can be cited independently in the future. For instructions see: http://journals.plos.org/plosone/s/submission-guidelines#loc-laboratory-protocols

We look forward to receiving your revised manuscript.

Kind regards,

Dr. Sakamuri V. Reddy

Academic Editor

PLOS ONE

Journal Requirements:

Additional Editor Comments (if provided):

The authors demonstrated protective effect of the extract from a tree P. floribundum Vog reduced inflammatory markers of periodontitis in rats including bone loss, cytokines, antioxidant and oxidative stress implicating the use in periodontal therapy. Title: please rephrase the title as “Protective effect of Platymiscium floribundum Vogin tree extract on periodontitis inflammation in rats”. Short title: “Effects of P. floribundum extract on periodontitis in rats”. Abstract-simplify introduction and methodology described; revise the conclusion sentence: “…..associated with periodontitis.” Methods-(122) Revises the subtitles: Determination of bone remodeling (line 150) SOD and CAT activity assessment. Fig.1B-Effect of P. floribundum (0.1; 1 or 10 mg/kg) on the macroscopic view (first column), histological aspects (second column, and Scanning Electron Microscopy (SEM) (third columm) of periodontiumlabel the panels. I suggest the authors to label the Fig.1B panels and show by arrows the bone loss observed under periodontitis condition and treatment conditions.

Reviewers' comments:

Reviewer's Responses to Questions

**Comments to the Author**

1. Is the manuscript technically sound, and do the data support the conclusions?

Reviewer #1: Yes

2. Has the statistical analysis been performed appropriately and rigorously? 

Reviewer #1: Yes

3. Have the authors made all data underlying the findings in their manuscript fully available?

Reviewer #1: Yes

4. Is the manuscript presented in an intelligible fashion and written in standard English?

Reviewer #1: Yes

5. Review Comments to the Author

Reviewer #1: Although the manuscript is interesting in the inhibitory effects of the extracts from Platymiscium Floribundum Vogin on periodontitis there are serious problems. The authors should be addressing them.

1) The paper indicated that the tree extract inhibited the inflammation and alveolar bone resorption. However, it is unclear how the inhibitory mechanism on bone resorption is. Is the extract directly or indirectly osteoclasts or osteoblasts? If the extract is mediated by anti-inflammation and resulting in the bone resorbing inhibition what cytokine is mainly inhibition? The authors should be speculated the inhibitory mechanism in discussion section in according to the known mechanism of the extract on anti-inflammation.

2) The authors should try to use the combination of some inhibitors or neutralized antibodies such as COX2 inhibitor, NOS inhibitor or anti-TNFalpha antibody with the extract in periodontitis model animal

3) p18, line297: Is FAT correct, not but TALP?

6. PLOS authors have the option to publish the peer review history of their article (what does this mean?). If published, this will include your full peer review and any attached files.

Reviewer #1: No

---

## [Author Response · Author response to Decision Letter 0]

16 Sep 2019

To Dr. Sakamuri V. Reddy

Academic Editor

PLOS ONE

 September 14th, 2019

Subject: Manuscript Reference Number: PONE-D-19-17659

Dear Dr. Sakamuri V. Reddy,

 Thank you very much for your letter with the reviewers’ comments on the manuscript mentioned above. 

 We tried to address all the comments raised and we really hope that our answers will clarify these points. The reviewers' comments are reproduced in BLACK; our responses are detailed below in BLUE. 

 We are grateful for your help in improving the quality of our study and we hope that this revised version of the manuscript is suitable for acceptance in PLOS ONE.

Yours sincerely,

Mirna Marques Bezerra

Author to Correspondence 

Manuscript Reference Number: PONE-D-19-17659

Additional Editor Comments

The authors demonstrated protective effect of the extract from a tree P. floribundum Vog reduced inflammatory markers of periodontitis in rats including bone loss, cytokines, antioxidant and oxidative stress implicating the use in periodontal therapy. 

Title: please rephrase the title as “Protective effect of Platymiscium floribundum Vog in tree extract on periodontitis inflammation in rats”. 

ANSWER: The title was rephrased. Please, see the revised version of the manuscript with tracked changes.

 Short title: “Effects of P. floribundum extract on periodontitis in rats”. 

ANSWER: The short title was rephrased. Please, see the revised version of the manuscript with tracked changes.

Abstract - simplify introduction and methodology described.

ANSWER: The introduction and methodology described in the Abstract were simplified. Please, see the revised version of the manuscript with tracked changes.

Revise the conclusion sentence: “…..associated with periodontitis.” 

ANSWER: The conclusion sentence was revised. Please, see the revised version of the manuscript with tracked changes (lines 475-476).

Methods - (122) Revises the subtitles: Determination of bone remodeling (line 150) SOD and CAT activity assessment. 

ANSWER: The subtitles were revised: Determination of bone remodeling (line 172); SOD and CAT activity assessment (line 200).

Fig.1B - Effect of P. floribundum (0.1; 1 or 10 mg/kg) on the macroscopic view (first column), histological aspects (second column, and Scanning Electron Microscopy (SEM) (third columm) of periodontium label the panels. I suggest the authors to label the Fig.1B panels and show by arrows the bone loss observed under periodontitis condition and treatment conditions.

ANSWER: Fig. 1B panels are labeled and we show with black arrows the bone loss observed under periodontitis condition and treatment conditions (Please, see the revised version of the manuscript with tracked changes).

Reviewers' comments:

Reviewer's Responses to Questions

Comments to the Author

1. Is the manuscript technically sound, and do the data support the conclusions?

Reviewer #1: Yes

2. Has the statistical analysis been performed appropriately and rigorously? 

Reviewer #1: Yes

3. Have the authors made all data underlying the findings in their manuscript fully available?

Reviewer #1: Yes

4. Is the manuscript presented in an intelligible fashion and written in standard English?

Reviewer #1: Yes

5. Review Comments to the Author

Reviewer #1: Although the manuscript is interesting in the inhibitory effects of the extracts from Platymiscium Floribundum Vog in on periodontitis there are serious problems. The authors should be addressing them.

ANSWER: We really appreciate the interest of the reviewer in our data. We tried to address all the comments raised. 

1) The paper indicated that the tree extract inhibited the inflammation and alveolar bone resorption. However, it is unclear how the inhibitory mechanism on bone resorption is. Is the extract directly or indirectly osteoclasts or osteoblasts? If the extract is mediated by anti-inflammation and resulting in the bone resorbing inhibition what cytokine is mainly inhibition? The authors should be speculated the inhibitory mechanism in discussion section in according to the known mechanism of the extract on anti-inflammation.

ANSWER: In order to make our answer clear enough we will answer this question considering the three issues raised by the Reviewer #1, as follows: 

(A) Is the extract directly or indirectly osteoclasts or osteoblasts? 

(B) If the extract is mediated by anti-inflammation and resulting in the bone resorbing inhibition what cytokine is mainly inhibition? 

(C) The authors should be speculated the inhibitory mechanism in discussion section in according to the known mechanism of the extract on anti-inflammation.

(A) Is the extract directly or indirectly osteoclasts or osteoblasts?

ANSWER: It has been shown that in the bone remodeling cycle, bone formation is coupled with bone resorption to maintain the bone health (Florencio-Silva et al., Biomed Res Int. (2015) 2015:421746). The success of this process involves a complex cross-talk between bone cells (osteoblasts and osteoclasts) and immune system cells (T lymphocytes, macrophages, and neutrophils) (Dar HY et al., Front. Biosci. (Landmark Ed). (2018) 23:464-492). During bone loss diseases, such as periodontitis, inflammation promotes a disruption in the balance between bone formation/bone degradation in towards of resorption (Hienz et al., J Immunol Res. 2015; 2015:615486). In response to inflammation-induced bone loss, neutrophils are usually the first cell type migrating to damage sites. Activated neutrophils can secrete a plethora of inflammatory mediators, such as, chemokines, cytokines, reactive oxygen species, prostaglandins, and small molecules (nitric oxide), which are able to act as immunomodulatory factors, then amplifying the inflammatory response, leading to the occurrence of bone loss (Amarasekara et al., J Immunol Res. (2015) 2015:832127; Domazetovic et al., Clin Cases Miner Bone Metab. (2017) 14(2): 209-216; Lisowska et al., Drug Des Devel Ther. (2018) 12:1753-1758; Kalyanaraman et al., Nitric Oxide. (2018) 76:62-70). On the other hand, anti-inflammatory agents, such as, P. floribundum may positively disturb this process in favour of bone formation, which may be beneficial during bone disorders related to inflammation, suggesting its use as a potential bone-protecting agent. 

 In the present study we tried to understand the underlying mechanisms of the anti-inflammatory action of P. floribundum by using an in vivo system to assess both soft and hard tissue destruction. Our research group has shown in last years that the inflammation-induced bone loss during periodontitis in rats is a complex process, which involves many players (Freitas, et al., Biomedicine & pharmacotherapy (2018) 98:863-872; Teixeira et al., Frontiers in Physiology (2017) 8:988; Sousa et al., Journal of Periodontology (2016) 87:1206-1216). Indeed, in the present study P. floribundum reduced bone loss and this action was due to its broad anti-inflammatory action including decreased MPO activity, TNF-α, IL1-β, IL-8/CINC-1, and PGE2 gingival levels, oxidative stress, and transcription of TNF-α, IL1-β, COX-2, iNOS, RANK and RANKL genes, while increasing both IL-10 gingival levels and bone alkaline phosphatase (BALP) serum levels. Considering our current data we can speculate that the inhibitory mechanism on bone resorption promoted by P. floribundum in the present study may be indirectly on both bone cells (osteoclasts and osteoblasts), and it might be related to the anti-inflammatory action of P. floribundum, including: reducing of both neutrophils infiltrate (evaluated by the levels of MPO), and the amount of pro-inflammatory mediators, while increasing the synthesis of IL-10, an anti-inflammatory cytokine.

 Therefore, understanding the communication between the immune systems and the skeletal is important for the discovery and development of novel anti-reabsortive drugs and it will clarify the known signal transduction pathways.

(B) If the extract is mediated by anti-inflammation and resulting in the bone resorbing inhibition what cytokine is mainly inhibition? 

 The network of inflammatory cytokines produced during inflammation induces an uncoupling of bone formation and resorption, resulting in significant bone loss (Amarasekara et al., J Immunol Res. (2015) 2015:832127). Ferreira et al (1993) (Br J Pharmacol. 110(3):1227-31) developed the concept that, in rats, inflammatory stimuli cause mechanical hypernociception by a well-defined sequential release of cytokines. These authors showed that among cytokines, TNF-α has a pivotal role in the development of inflammatory hyperalgesia. Indeed, the first cytokine released is TNF-α, which triggers the release of IL-6/IL-1β and cytokine-induced neutrophil chemoattractant-1 (CINC-1) (human IL-8-related chemokine), which is responsible for stimulation of the synthesis of prostaglandins (Cunha et al., Br. J. Pharmacol. (1991) 104: 765–767; Cunha et al., Br. J. Pharmacol. (1992) 107: 660–664; Lorenzetti et al., Eur. Cytokine Network (2002)13: 456–461).

 During periodontitis cytokine involvement is well known, and these molecules, particularly tumor necrosis factor-α (TNF-α) and interleukin-1β (IL-1β) may amplify the inflammatory response, causing tissue destruction and bone loss. TNF-α and IL-1β are the first to appear in periodontitis pathways and they are associated with inflammatory cell migration (Hoare et al., Mediators Inflamm. (2019) 2019:1029857). Also, the release of TNF-α and IL-1β during periodontitis activates osteoblasts and osteoclasts to produce COX2-mediated prostaglandin E2 (PGE2), triggering bone resorption (Yu et al., J Periodontal Res. (2016) 51(1):38-49). Further, preclinical and clinical trials have proven the protective effect of both TNF-α and IL-1β inhibitors on periodontitis (Lima et al., J. Periodontol. (2004) 75: 156–162; Holmlund et al., J Clin. Periodontol. (2004) 31: 475–482). Therefore, the inhibition of cytokine production or action appears to constitute a real target for a new therapeutic approach to control the inflammatory bone diseases such as periodontitis.

 Thus, considering the previous data showing the central role of TNF-α during inflammatory response by activating the cascade of cytokine release, which consequently amplifies the degree of inflammation, and its involvement during periodontitis, we may speculate that TNF-α could be the main cytokine involved in the anti-inflammatory efficacy of P. floribundum.

(C) The authors should be speculated the inhibitory mechanism in discussion section in according to the known mechanism of the extract on anti-inflammation.

Answer: Although P. floribundum is popularly used as an anti-inflammatory [19], the literature has few studies concerning its mechanism of action. In this regard, phytochemical investigations demonstrated the induction of apoptosis by flavonoids from P. floribundum in HL-60 human leukemia cells (reference number 6 - Militão et al., Life Sci. (2006); 78 (20): 2409-2417). Further, it was reported P. floribundum antifungal, DNA-damaging and anticholinesterasic activities (reference number 7 - Cardoso-Lopes Rev. Bras. Farmacogn. 2008; 18: 655-660). The use of natural products in the management of periodontitis still lacks preclinical and clinical studies that can prove its efficacy and safety. P. floribundum was chosen to be studied because it has been used in the Brazilian Northeast in folk medicine as an anti-inflammatory agent and no previous pharmacology study was reported yet. Indeed, the present study is the first demonstration of both efficacy and safety of P. floribundum as an anti-inflammatory agent. In keeping with this assumption, we included a sentence in the discussion section mentioning this issue (Please, see the revised version of the manuscript with tracked changes, lines 391-405). 

2) The authors should try to use the combination of some inhibitors or neutralized antibodies such as COX2 inhibitor, NOS inhibitor or anti-TNFalpha antibody with the extract in periodontitis model animal.

ANSWER: We really appreciate the interest of the reviewer in our protocol elaboration. The main idea of our research group is to develop a natural product, which may provide a better risk/benefit ratio in the treatment of periodontitis. Herein, we carefully draw a suggestion that P. floribundum has a regulatory role in inflammation-induced bone loss during periodontitis, reducing the main inflammatory mediators that orchestrates bone loss. Indeed, P. floribundum decreased TNF-α and PGE2 gingival levels, and transcription of TNF-α, COX-2, and iNOS genes. Therefore, considering these data, and keeping in mind the 3R Principle (Russel and Burch, 1959), which suggests the reduction of the animal numbers whenever possible, we did not consider the possibility of use the combination of some inhibitors or neutralized antibodies such as COX2 inhibitor, NOS inhibitor or anti-TNFalpha antibody with the extract in periodontitis model animal.

3) p18, line 297: Is FAT correct, not but TALP?

ANSWER: The acronym FAT is not correct and it was replaced by the correct acronym TALP. Please, see the revised version of the manuscript with tracked changes (line 349).

---

## [Decision Letter · Decision Letter 1]

30 Sep 2019

Protective effect of Platymiscium floribundum Vog in tree extract on periodontitis inflammation in rats.

PONE-D-19-17659R1

Dear Dr. Freire,

We are pleased to inform you that your manuscript has been judged scientifically suitable for publication and will be formally accepted for publication once it complies with all outstanding technical requirements.

With kind regards,

Dr. Sakamuri V. Reddy

Academic Editor

PLOS ONE

Additional Editor Comments (optional):

Reviewers' comments:

Reviewer's Responses to Questions

**Comments to the Author**

1. If the authors have adequately addressed your comments raised in a previous round of review and you feel that this manuscript is now acceptable for publication, you may indicate that here to bypass the “Comments to the Author” section, enter your conflict of interest statement in the “Confidential to Editor” section, and submit your "Accept" recommendation.

Reviewer #1: All comments have been addressed

2. Is the manuscript technically sound, and do the data support the conclusions?

Reviewer #1: Yes

3. Has the statistical analysis been performed appropriately and rigorously? 

Reviewer #1: Yes

4. Have the authors made all data underlying the findings in their manuscript fully available?

Reviewer #1: Yes

5. Is the manuscript presented in an intelligible fashion and written in standard English?

Reviewer #1: Yes

6. Review Comments to the Author

Reviewer #1: The authors have almost addressed and revised the pointed-out problems. The reviewer recommend it.

7. PLOS authors have the option to publish the peer review history of their article (what does this mean?). If published, this will include your full peer review and any attached files.

Reviewer #1: No

---

## [Editor Report · Acceptance letter]

23 Oct 2019

PONE-D-19-17659R1 

Protective effect of *Platymiscium floribundum* Vog in tree extract on periodontitis inflammation in rats. 

Dear Dr. Freire:

I am pleased to inform you that your manuscript has been deemed suitable for publication in PLOS ONE. Congratulations! Your manuscript is now with our production department. 

With kind regards,

on behalf of

Dr. Sakamuri V. Reddy 

Academic Editor

PLOS ONE